# Comparison of the Effects of Aerobic versus Resistance Exercise on the Autonomic Nervous System in Middle-Aged Women: A Randomized Controlled Study

**DOI:** 10.3390/ijerph19159156

**Published:** 2022-07-27

**Authors:** Chae Kwan Lee, Jae-Hoon Lee, Min-Seong Ha

**Affiliations:** 1Department of Physical Therapy, Catholic University of Pusan, 57 Oryundae-ro, Geumjeong-gu, Pusan 46252, Korea; cklegend@naver.com; 2Department of Sports Science, University of Seoul, 163 Seoulsiripdae-ro, Dongdaemun-gu, Seoul 02504, Korea; leejh0923@gmail.com; 3Department of Sports Culture, College of the Arts, Dongguk University-Seoul, 30 Pildong-ro 1-gil, Jung-gu, Seoul 04620, Korea

**Keywords:** autonomic nervous system, resistance training, aerobic exercise, sympathetic nerve, parasympathetic nerve, middle-aged women

## Abstract

This study was conducted to investigate the changes in the autonomic nervous system in middle-aged women induced by aerobic and resistance exercise. A randomized controlled design was adopted; 22 premenopausal middle-aged women were divided into the resistance training and aerobic exercise groups (*n* = 11 each). Each group followed a specific 60 min exercise program three times a week for 12 weeks. The participants’ heart rate variability (HRV) was measured to analyze the low(LF)- and high-frequency (HF) activity, and the LF/HF ratio was calculated to examine the autonomic nervous system’s activities. A repeated-measures analysis of variance was used to analyze the effects of resistance and aerobic exercise. LF activity significantly increased in both the resistance training (*p* < 0.001) and aerobic exercise (*p* < 0.5) groups, indicating a significant variation according to time effect. HF activity was significantly increased only in resistance training (*p* < 0.001) with a significant variation in time (*p* < 0.001) and an interaction effect (*p* < 0.01). The LF/HF ratio did not vary significantly in either group. The findings in this study suggest that both aerobic exercise and resistance training were effective for sympathetic nerve activities in middle-aged women and that the effects on the sympathetic and parasympathetic activities were greater for resistance training.

## 1. Introduction

Middle-aged women undergo sudden hormonal changes resulting in a decrease in muscle mass and a rise in abdominal fat content, which can lead to obesity and increased susceptibility to chronic degenerative diseases [1,2]. In addition, reduced physical activity and fitness cause physiological and psychological stress, including depression and anxiety. Middle-aged women experiencing such diverse stress negatively impacts the body, while continuous stress induces more varied and severe problems as aging progresses.

Stress caused by diverse sources disrupts the body’s homeostasis leading to abnormalities including mental or physical fatigue, insomnia, lassitude, obesity, and cardiac tamponade [3]. The autonomic nervous system plays a critical role in controlling homeostasis, consisting of low-frequency (LF) power for sympathetic activities and high-frequency (HF) power for parasympathetic activities that are reduced with aging [4]. Inhibition of the autonomic nervous system function can cause various adverse effects on the body, and menopausal women show a difference in sympathetic-vagus activation [5,6]. It is important to control the autonomic nervous system prior to these changes, and women who lead an active life show lower autonomic nervous system activity than women who do not [6]. There is insufficient information on the autonomic nervous system responses in premenopausal middle-aged women.

The autonomic nervous system controls the blood vessels, internal organs, and endocrine glands that cannot be consciously regulated. The system thus plays a crucial function in maintaining the body’s internal environment and homeostasis against the external environment [7]. This system consists of the sympathetic and parasympathetic nerves in antagonism; whereby the increase in the former increases cardiac pressure, stroke volume, and pulse rate as well as muscle tension [8,9]. When an increase in sympathetic nerve activity is accompanied by decreased parasympathetic nerve activity to cause impaired function, the autonomic nervous system loses its balance, causing a notable increase in the risk of cardiovascular disease [3,10].

Regular physical exercise induces numerous health benefits, including cardiopulmonary fitness, blood pressure and lipids, and psychological health to improve individuals’ general health [5,6]. It is also known to assist in preventing and improving various diseases such as cardiovascular disease and metabolic syndrome and their related mortalities [11,12,13,14,15]. Enhanced cardiopulmonary fitness entails improvements in heart rate and stroke volume regulated via the central command, exercise pressor reflex, and active arterial baroreflex, which are associated with the control of the autonomic nervous system [9].

Aerobic exercise modulates autonomic nerve activity and improves imbalances of hypersympathetic or decreased parasympathetic nerves [16]. Transient aerobic exercise promotes sympathetic activity so that the heart rate increases, and depending on the intensity, vagal activity is suppressed [8,17]. This implies a potential effect of exercise on the activities of autonomic nerves. The reported long-term effects of aerobic exercise include reduced blood pressure due to changes induced via baroreflex sensitivity [18,19] and reduced heart rate due to the enhanced regulation of vagal activity [20]. In addition, a meta-analysis of the effects of progressive resistance training on blood pressure in adults showed that resistance training could reduce both systolic and diastolic blood pressure at rest [21]. Conversely, Kingsley et al. [22] reviewed 18 studies on short- and long-term resistance training to identify the effects on the autonomic nervous system and reported a lack of long-term effects despite the short-term effects, which increase sympathetic nerve activity and decrease parasympathetic nerve activity.

While studies have continuously investigated the effects of exercise on the autonomic nervous system, most focused on the effects of a single exercise type or a combined exercise program containing an aerobic exercise to report contrasting results according to characteristics of participants and type of interventions. Furthermore, no prospective study has yet confirmed the effects of long-term exercise on the autonomic nervous system in middle-aged women. Thus, to determine which exercise type positively affects the autonomic nervous system in middle-aged women, this study investigated the independent effects of aerobic exercise and resistance training. This study aimed to identify the type of exercise that effectively controls the autonomic nervous system in middle-aged women.

## 2. Materials and Methods

### 2.1. Participants

The participants in this study were all premenopausal middle-aged women. The sample size, estimated using G-power version 3.1 for Windows (Kiel University, Kiel, Germany) at effect size: 0.25 (default), significance: 0.05, and power: 0.60, was *n* = 21. Considering a drop-out rate, 30 participants were recruited. Before the experiment, all participants were informed of and showed an understanding of the study’s goals and purpose. Written signed consent was collected from each participant, while the study protocol was approved by the Institutional Review Board at the Catholic University of Pusan and was performed in compliance with the Helsinki Declaration and ethical research principles (CUPIRB/2018_038). Individuals with no injury were included from this study based on the physician’s consultation, interview, and physical examination, and those with nerve or musculoskeletal disorders that could affect the experiment were excluded. The final number of participants was 22, and the participants were randomized into the resistance training group (*n* = 11) and the aerobic exercise group (*n* = 11; Table 1; Figure 1).

### 2.2. Measurements

To control any other factors that may affect the autonomic nervous system in this study, the laboratory was blocked from noise and light, and the temperature and humidity were maintained at 21–23 °C and 50%, respectively. The participants were prohibited from exercising and drinking alcohol for 1 day before the experiment and were prohibited from eating, smoking, and consuming caffeinated drinks for 3 h before the experiment. Autonomic nervous system measurements were taken at 9–11 a.m. using the uBio Clip v70 (Biosense creative, Seoul, Korea) after 10 min of resting by sitting before a desk.

The heart rate variability (HRV) indicates the variation in the R–R interval between one cardiac cycle and the next [23,24]. Analyzing the changes from one heartbeat to the next provides information regarding the autonomic nervous system’s sympathetic and parasympathetic activities. The HRV allows specific and complex analyses of the HRV so that the role of the autonomic nervous system in the heart to allow noninvasive measurement and diagnosis [25]. To measure the HRV, the participant wore the clip on their left forefinger positioned at the level of the heart and maintained for 2 min 30 s, and the optical sensor was used noninvasively. The autonomic nerve activities were evaluated based on the data extracted from the pulse wave analyzer. For frequency range, the LF of 0.04–0.15 Hz indicating sympathetic activity, the HF of 0.15–0.4 Hz indicating parasympathetic activity, and the LF/HF ratio indicating the autonomic nervous system’s balance were analyzed.

### 2.3. Heart Rate Variability Measurement Device

HRV was calculated using the uBioMacpa Vital system (uBioClip v70). uBio-Clip v70 is a medical device certified by the Ministry of Food and Drug Safety (certification number KTL-AA-110,168). The system can identify stress state by measuring heart rate through HRV analysis of capillaries. Stress measurement using the state-of-the-art device, uBioClip v70, is based on criteria for each item such as age-specific sympathetic nerve activity, parasympathetic nerve activity, autonomic balance, average pulse rate, and standard deviation. Pulse rate and mean deviation according to HRV analysis criteria of the North American Society of Pacing and Electrophysiology and European Society of Cardiology [26]. Autonomic nerve analysis and evaluation were performed using these criteria. According to the measurement principle of uBioClip v70, light is emitted to the surrounding capillaries using a light emitting diode (LED) light source and a sensor for analysis, and the absorbed and reflected light is converted into a signal. The rate of absorption of light represents the maximum systolic blood pressure and the minimum diastolic blood pressure. In this study, the HRV index (unit: ms) is the calculated standard deviation of all normal-to-normal (NN) intervals (SDNN).

### 2.4. Exercise Programs

Resistance exercise and aerobic exercise programs in our study were conducted in accordance with the ACSM guidelines. Unlike aerobic exercise, anaerobic exercise was not based on time, but the set and number of repetitions [26]. The resistance training program was based on a previous study of middle-aged women by Park (2008) [27]. The intensity was measured using the one repetition maximum (1RM) method (1RM = lifted weight [kg]/1.0278 − [repeated frequency × 0.0278]) [28]. Three sets were performed with 10–15 repetitions per set; 65% of 1RM at 1–4 week, 70% of 1RM at 5–8 week, and 75% of 1RM at 9–12 week. The details of the resistance training program are given in Table 2.

The aerobic exercise program was based on the American College of Sports Medicine guidelines. The intensity was set to 50–60% heart rate reserve (HRR) at W1–4, 60–70% HRR at W5–8, and 70–75% HRR at W9–12. The aerobic exercise intensity was measured using the Polar, a wristwatch-type heart rate device (Polar RCX, Kempele, Finland) that measures HRV to indicate the HRR. The details of the aerobic exercise program are given in Table 3.

### 2.5. Data Analysis

The data collected in this study were analyzed using IBM SPSS Ver 27. The mean and standard deviation were obtained for each measured variable. To determine the effects of the 12-week aerobic exercise and resistance training on the sympathetic and parasympathetic activities, 2 × 2 two-way repeated-measures analysis of variance was used with treatment (resistance training and aerobic exercise) and time (pre- and post-training) set as independent variables. The Bonferroni test was used for post-hoc testing, and the effect size (Cohen’s d) for the pretest–posttest analysis was expressed as the mean variation [29]. The level of significance was set to *p* < 0.05.

## 3. Results

### 3.1. LF Activity

Figure 2 shows the result of analyzing the effects of 12-week resistance training and aerobic exercise on the LF values to indicate sympathetic nerve activities. For within-group variation, the LF values significantly increased from 6.13 ± 0.96 to 7.78 ± 1.41 ms^2^ in the resistance training group (*p* < 0.001) and from 6.80 ± 1.04 to 7.54 ± 1.15 ms^2^ in the aerobic exercise group (*p* < 0.05). Both groups showed a significant increase in LF values, which indicated a significant change with time, while no significant between-group variation or interaction was found.

### 3.2. HF Activity

Figure 3 shows the result of analyzing the effects of 12 week resistance training and aerobic exercise on the HF values as an indicator of parasympathetic nerve activities. For within-group variation, the HF significantly increased from 5.06 ± 0.71 to 6.58 ± 1.11 ms^2^ in the resistance training group (*p* < 0.001), whereas the increase in the aerobic exercise group was from 6.10 ± 0.74 to 6.41 ± 1.13 ms^2^, showing no statistical significance. Only the resistance training group showed a significant increase in HF values, which indicated a significant change with time (*p* < 0.001) and interaction (*p* < 0.01).

### 3.3. LF/HF Ratio

Figure 4 shows the result of analyzing the effects of 12-week resistance training and aerobic exercise on the LF/HF ratio as an indicator of the autonomic nervous system’s balance. No significant change was found in the LF/HF ratio in the resistance training and aerobic exercise groups, and no significant time-dependent or between-group variations or interactions were observed. The LF/HF ratio decreased from 1.12 ± 0.11 to 1.19 ± 0.11 ms^2^ in the resistance exercise group and increased from 1.12 ± 0.13 to 1.19 ± 0.11 ms^2^ in the aerobic exercise group.

## 4. Discussion

To determine the effects of aerobic exercise and resistance training on the autonomic nervous system in middle-aged women, 22 participants were recruited and divided into the resistance training (*n* = 11) and aerobic exercise (*n* = 11) groups. The effects on the LF and HF activity and LF/HF ratio were analyzed, and the results showed that resistance training could increase LF activity to a greater degree than aerobic exercise, while only resistance training significantly increased HF activity.

Exercise is generally known to control heart rate and cardiac output by improving the autonomic nervous system activity and restoring the balance (LF/HF ratio) between the sympathetic and autonomic nervous systems. In a study on mice, similar results were obtained after training to verify the effect of exercise in regulating the autonomic nervous system [16]. In the results of this study, LF was significantly increased with time, although there was no difference between the resistance exercise group and the aerobic exercise group. This result supports those of previous studies, including one regarding menopausal women and the effects of a 6-month exercise program, where the level of 4 kcal/kg per week or above, suggested by the National Institutes of Health, could increase LF activity to induce a change in HRV [30] and another regarding menopausal women and the effects of an 8-week moderate intensity exercise program in increasing the HRV, LF, HF [31].

The effect of exercise on increasing sympathetic nervous activity may be attributed to the increased heart rate for which the autonomic nervous system was promoted to control the heart function. Notably, the increase in sympathetic activity was in line with the exercise intensity, while the heart rate was controlled by inhibiting vagal activity [8,17]. As aging progresses, the activity of the autonomic nervous system decreases [4]. In this study, the reduction in LF activity due to aging in middle-aged women may have been increased by the 12-week aerobic exercise and resistance training programs.

A significant increase in HF activity was found only in the resistance training group. In a review of resistance training regarding its effects on the autonomic nervous system, no long-term effect was found in younger adults, middle-aged men, or middle-aged women with mild hypertension [22]. However, in a study on middle-aged women with fibromyalgia, exercise intensity is gradually increased from 50% to 80% of 1RM for 16 weeks resistance training was found to increase the sympathetic and parasympathetic activities in support of these results [32]. The result also agreed with Jurca et al. [31] who studied female older adults who performed exercises of three different intensities, where the high-intensity exercise effectively increased parasympathetic activity, and with a study on younger adults who performed 10-week maximum-intensity weight training, which showed a positive effect of exercise of vagal control [33].

No significant variation was observed HF in this study for the aerobic exercise group. Hautala et al. [20] suggested that a relatively lower intensity of aerobic exercise could effectively regulate vagal activity. The results of this study can be used as basic data to find the effective intensity of exercise to control the autonomic nervous system in middle-aged women who have not yet taken a prospective approach.

The LF/HF ratio did not vary significantly in either group. In healthy adults, the HRV should not exceed the following levels: LF = 7.31, HF = 6.95, and LF/HF = 1.98 [34]. Chronic sympathetic nerve hyperactivity increases stroke volume, heart rate, and peripheral resistance to increase the stress on the heart and cause a hemodynamic overload, increasing the risk of cardiovascular disease [3,35]. A low HRV indicates an unhealthy cardiovascular state associated with the high mortality of sudden heart diseases and mortality due to myocardial infarction, angina, and coronary artery disease [36]. In another previous study regarding sedentary male workers, a cycle aerobic exercise positively affected HRV after 12 weeks [19]. In a study on menopausal women, the group that performed aerobic exercise for ≥2 years showed higher HRVs [37].

However, although a 10-week aerobic exercise program led to reduced blood pressure and heart rate in adults aged ≥ 55 years, no significant change in the autonomic nervous system’s balance was observed [19]. Similarly, in this study, the lack of significant variation in the LF/HF ratio showed that aerobic exercise and resistance training did not disturb the balance between the sympathetic and parasympathetic nerves’ activities, while a significant increase was found for LF and HF activity in the resistance training group and LF activity in the aerobic exercise group. 

In a systematic literature review and meta-analysis conducted to verify the effect of resistance training on heart rate variability, most studies on healthy adults did not show that resistance exercise changed heart rate variability. However, it improved in subjects with disease or impaired cardiac autonomic control [38]. Based on the findings in this study, resistance training could increase LF and HF activity, and aerobic exercise could increase LF activity. This suggested that, as the HRV was increased by resistance training and aerobic exercise, both exercise types positively affected participants’ health. Notably, as the parasympathetic nerve activity was increased to a greater degree by resistance training than aerobic exercise, resistance training had a more positive effect in middle-aged women with reduced parasympathetic activity.

Nevertheless, there are several limitations to the generalizability of the results in this study. First, as the sole focus of the study was on middle-aged women, it is difficult to generalize the results to men or women of other age groups. Second, the LF and HF activities and LF/HF ratio were analyzed to examine the effects on the autonomic nervous system, and no other indicators were investigated. In addition, sympathetic or parasympathetic nerve activity could not be obtained directly using the results of this study. Furthermore, hematological analysis could not be performed and only indicators of integrated regulation could be presented. Thus, a follow-up study should apply a greater variety of exercise intensity, frequency, and types to further clarify the relationship between the autonomic nervous system and exercise. Further studies should also involve more complex analyses of those indicators and others regarding (blood, etc.) the autonomic nervous system. 

## 5. Conclusions

The autonomic nervous system modulates various physiological functions in the human body, and is thus a critical factor in middle-aged women undergoing numerous physiological changes, such as a reduction in muscle mass. This study thus analyzed the changes in the autonomic nervous system’s response in middle-aged women after aerobic exercise and resistance training, and we drew the following conclusions: LF activity as an indicator of sympathetic nerve activity showed a significant increase in aerobic exercise and resistance training groups. As an indicator of parasympathetic nerve activity, HF activity showed a significant increase in the resistance training group only. No significant variation was shown in the aerobic exercise group, but a significant increase was found in the resistance training group with a significant interaction effect.

These results suggest that exercise is essential to increasing the autonomic nervous system’s balance in middle-aged women. While both aerobic exercise and resistance training effectively increase the sympathetic nerve activity of middle-aged women, the effect of resistance training is more significant on sympathetic and parasympathetic activities together.

## Figures and Tables

**Figure 1 ijerph-19-09156-f001:**
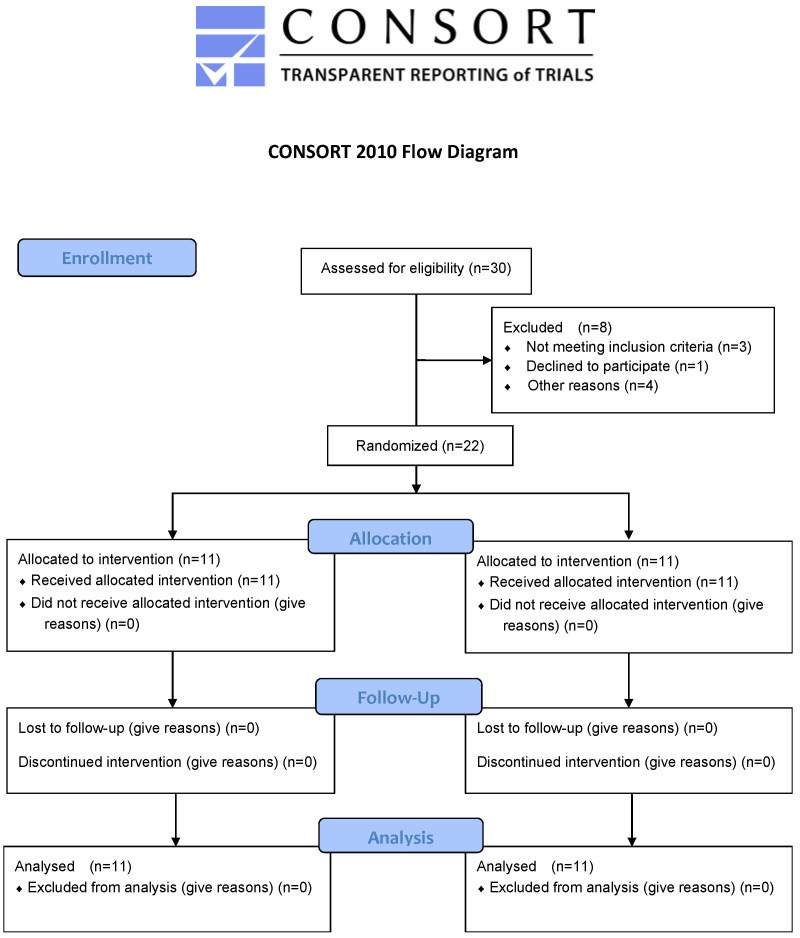
CONSORT flow diagram for the individual randomized, controlled trial.

**Figure 2 ijerph-19-09156-f002:**
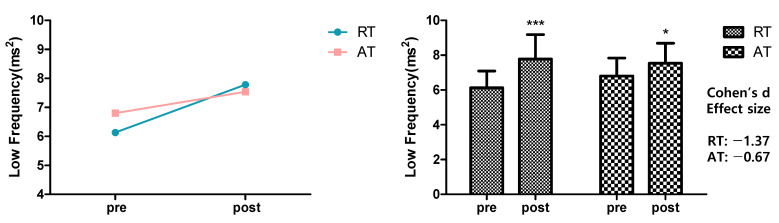
Effects of 12-week resistance training and aerobic training on Low Frequency. LF significantly increased in the RT (*** *p* < 0.001), AT (* *p* < 0.05) compared with the pre-test. However, LF showed no significant interaction effect. The data presented were mean ± SD. * *p* < 0.05, *** *p* < 0.001; before and after significant. Effect size range: |0.20| ≤ small < |0.50| < medium < |0.80| ≤ large. RT = resistance training, AT = aerobic training, LF = low frequency.

**Figure 3 ijerph-19-09156-f003:**
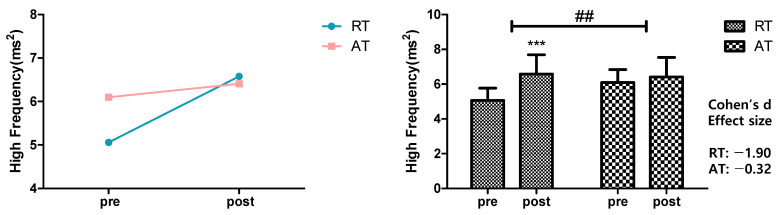
Effects of 12-week resistance training and aerobic training on Low Frequency. HF has significantly increased in the RT (*** *p* < 0.001) compared to the pre-test. HF showed a significant interaction effect (## *p* < 0.01). The data presented were mean ± SD. *** *p* < 0.001; before and after significant. ## *p* < 0.01; interaction significant. Effect size range: |0.20| ≤ small < |0.50| < medium < |0.80| ≤ large. RT = resistance training, AT = aerobic training, HF = high frequency.

**Figure 4 ijerph-19-09156-f004:**
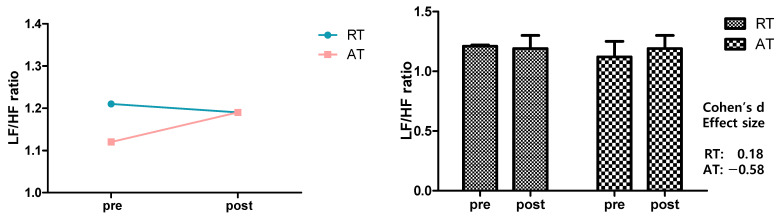
Effects of 12-week resistance training and aerobic training on Low Frequency/High Frequency ratio. LF/HF showed no significant changes in any group. The data presented were mean ± SD. Effect size range: |0.20| ≤ small < |0.50| < medium < |0.80| ≤ large. RT = resistance training, AT = aerobic training, LF = low frequency, HF = high frequency.

**Table 1 ijerph-19-09156-t001:** General characteristics of the study subjects.

	Variables	Age (years)	Height (cm)	Weight (kg)	BMI (kg/m^2^)
Group					
RT (*n* = 11)		50.4 ± 1.6	159.3 ± 2.8	63.6 ± 3.2	22.4 ± 2.3
AT (*n* = 11)		49.1 ± 1.5	157.5 ± 3.9	61.1 ± 4.7	22.1 ± 3.0

Values are presented as M ± SD. RT: resistance training, AT: aerobic training.

**Table 2 ijerph-19-09156-t002:** Resistance training program.

Week	Order	Exercise	Intensity		Frequency
	Warn-Up (10 min)	Static Stretching			
1–4	Main exercise (40 min)	1. Bench press 2. Leg press 3. Lat pull down 4. Leg curl 5. Shoulder Press 6. Leg extension 7. Biceps curl 8. Triceps extension	3 set	65% 1RM	3 times/week
5–8			3 set	70% 1RM	
9–12			3 set	75% 1RM	
	Cool-down (10 min)	Static stretching			

**Table 3 ijerph-19-09156-t003:** Aerobic exercise program.

Week	Order	Exercise	Intensity	Frequency
	Warn-Up (10 min)	Static Stretching		
1–4	Main exercise (40 min)	Treadmill walking or jogging	50–60% HRR	3 times/week
5–8			60–70% HRR	
9–12			70–75% HRR	
	Cool-down (10 min)	Static stretching		

## Data Availability

The authors declare that all data and materials are available to be shared on a formal request.

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
