# Peer review of "Comparison of the Effects of Aerobic versus Resistance Exercise on the Autonomic Nervous System in Middle-Aged Women: A Randomized Controlled Study"

_ijerph, 2022, doi:10.3390/ijerph19159156_

Round 1

Reviewer 1 Report

Thank you for making the changes to the manuscript, it is much improved. In my opinion, there is still areas that could be improved but it is now at a standard that is acceptable for publication.

Thank you for including the CONSORT diagram, however you should check this as it is suggesting that you lost all participants to follow-up, which I am not sure is actually the case.

Author Response

Response to Reviewers

Manuscript Title: Comparison of the effects of aerobic versus resistance exercise on the autonomic nervous system in middle-aged women: A randomized controlled study

Editorial Board Member,

We appreciate the positive comments from the reviewers. Based on your comments, our manuscript was finally revised. We eagerly look forward to publishing the revised manuscript in the International Journal of Environmental Research and Public Health.

To Reviewers:

We thank two reviewers for their positive suggestions on our manuscript. We carefully read each reviewer's comments and revised the manuscript accordingly. Each comment is outlined below, point-by-point. Therefore, we believe that the revised version of the manuscript is a significant improvement over the initial submission.

Reviewer 1

Comments and Suggestions for Authors

Thank you for making the changes to the manuscript, it is much improved. In my opinion, there is still areas that could be improved but it is now at a standard that is acceptable for publication.

Comment: Thank you for including the CONSORT diagram, however you should check this as it is suggesting that you lost all participants to follow-up, which I am not sure is actually the case.

Author’s response: Thanks for your thoughtful comment. First of all, thank you for your positive evaluation of our research paper. We corrected after confirming that there was an error in the follow-up part of CONSORT diagran. Please check again.

Thank you for your kind and positive review. We have tried to apply your comments carefully. And English editing was done by a native English speaker. We look forward to your positive judgment.

Reviewer 2 Report

Authors did not improve sufficiently the revised manuscript.

As an example, text regarding study design , methods to obtain indices of autonomic regulation, and their variation (SD) (figures) have not been clarified sufficiently.  

The meaning of: "improving the autonomic nervous system activity and restoring the balance between the sympathetic and autonomic nervous systems"  is still unclear.

aging progresses, the activity of the autonomic nervous system decreases (needs a relevant quote, usually the opposite is found). Moreover, women have lower LF power than men

implies that spectral powers measure sympathetic activity.

Author Response

Response to Reviewers

Manuscript Title: Comparison of the effects of aerobic versus resistance exercise on the autonomic nervous system in middle-aged women: A randomized controlled study

Editorial Board Member,

We appreciate the constructive comments from two reviewers. Based on reviewer’s comments, our manuscript was finally revised. We eagerly look forward to publishing the revised manuscript in the International Journal of Environmental Research and Public Health.

To Reviewers:

We thank the reviewers for their sharp suggestions on our manuscript. We read the reviewers' comments carefully and revised the manuscript accordingly. Each comment is summarized below. Therefore, we believe that the revised version of the manuscript is a significant improvement over the initial submission.

Reviewer 2

Comments and Suggestions for Authors

Authors did not improve sufficiently the revised manuscript.

Comment: As an example, text regarding study design, methods to obtain indices of autonomic regulation, and their variation (SD) (figures) have not been clarified sufficiently.  

Author’s response: Thanks for your comment. We have additionally described the study design and methodology used by the reviewers. The autonomic nervous system was also described in detail. If there is something missing, it would be helpful if you could provide us with a more detailed comment. The lack of explanation of SD in figures seems to be because the standard deviation in the first graph is missing from our results. However, in the second graph, the standard deviation is already displayed, so it is shown in the second graph. We corrected and rewritten the graph of LF/HF ratio because it overlapped with HF, and there was no information about the result value in this part.

Comment: The meaning of: "improving the autonomic nervous system activity and restoring the balance between the sympathetic and autonomic nervous systems"  is still unclear.

Author’s response: We are sorry that you were not satisfied with our response to your comment. We have made corrections once again in the areas you pointed out. In particular, the part explaining the difference between men and women was deleted after checking. We look forward to your positive judgment.

Comment: aging progresses, the activity of the autonomic nervous system decreases (needs a relevant quote, usually the opposite is found). Moreover, women have lower LF power than men implies that spectral powers measure sympathetic activity.

Author’s response: We would like to apologize for not understanding you. We have corrected not only the parts you pointed out, but also the parts that may be misunderstood. In particular, we tried to reduce misunderstandings by using the expression "increased" in the expression "improved". In addition, specific explanations have been added.

Thank you for your kind and sharp review. We have tried to apply your comments carefully. And the English editing was done by a native English speaker. We hope that our responses to your comments have made you feel at ease. We look forward to your positive judgment.

This manuscript is a resubmission of an earlier submission. The following is a list of the peer review reports and author responses from that submission.

Round 1

Reviewer 1 Report

The manuscript ijerph-1712044 by Lee CK et al addresses a topic of physiopathological relevance.   Indeed the dynamics of cardiovascular neural regulation in response to exercise is of paramount importance both physiologically and clinically.  According to the point of view the description of autonomic aspects may change substantially.  Among the multiple viewpoints the difference between dynamic and static activity is critical, although relatively few studies address this latter aspect.

Accordingly the choice of Authors has to be praised.  However the design (two small parallel groups) is not the best to assess the difference of aerobic vs resistance exercise.  A cross over design may be preferable.

In addition selecting only middle aged women further limits the potential interest of possible readers.

Finally the description of the background is too general: both physiologically, see Introduction  from Para 3, and  methodologically.  Moreover there are some aspects that require rewriting: e.g. “aerobic exercise decreases sympathetic …activity” cannot be accepted.  Probably Authors intended Bouts of exercise (or the like).

Moreover I did not really understood how Authors plan to “identify the type of exercise that …controls the ANS in middle aged women”

Furthermore the Methods should indicate what Authors are measuring and how they are using this signal to derive information on autonomic regulation.  By the way it is not easy to find information on uBioClip v70 and related software.

Finally it is not possible to obtain non invasively sympathetic  or parasympathetic “activity”, but only markers or indices of integrated regulation.

Regarding results the pre-post plots should better indicate the dispersion (SD)  around mean values.

Discussion also requires some rewriting.  E.g. exercise DOES NOT decrease sympathetic activity...(2nd para)

The following Para states: exercise increasing sympathetic nervous activity

Authors indicate paradoxically a number of important improvements or critical aspects:   focusing on hemodynamics,  considering only middle aged women, changes in muscle mass, more complex analysis, more extensive evaluation of exercise properties.

Reviewer 2 Report

The authors have attempted to determine which type of exercise more effectively controlled the autonomic nervous system in middle-aged women. To do so, they designed a randomised trial comparing resistance training and aerobic training for 12 weeks.

The introduction gave a nice overview of the autonomic system and how exercise might affect this, but it did not build a rationale for how different types of exercise or intensities of exercise might provide a different autonomic response. A rationale that logically leads the reader to the aim should be provided.

The methods provided an overview of the participant demographics. Further information about the method for measuring R-R intervals should be provided. The uBio Clip v70 is a non-traditional way of measuring this and I am unsure if the validity of this method has been determined? The validity should be stated or identified why this method was chosen over other valid methods.

More detail of the exercise programs should also be provided. I would recommend the CERT guidelines to the authors. More importantly, why where the exercise programs that were implemented chosen? This also feeds back into the lack of rationale and variations in intensity or work duration. Do the authors expect that different 'work' times (40 min for aerobic, and I am only guessing 20 min for resistance - assuming a 60 sec rest between sets). The authors should provide more rationale for the chosen exercise prescriptions to convince the reader that a valid comparison is being made.

The results should be revised to avoid repetition in the reporting. The authors should also provide a CONSORT diagram and report this trial according to the CONSORT statement.

While the discussion provided some comparison to previous literature there was no attempt made to try and explain the findings. Here again, I encourage the authors to consider trying to explain the effects of exercise intensity and the variations the prescription.

I would also encourage the authors to consider practical applications and revise the conclusion to be more conclusive and less of a discussion.